# Extreme Food Insecurity and Malnutrition in Haiti: Findings from a Population-Based Cohort in Port-au-Prince, Haiti

**DOI:** 10.3390/nu14224854

**Published:** 2022-11-17

**Authors:** Rehana Rasul, Vanessa Rouzier, Rodney Sufra, Lily D. Yan, Inddy Joseph, Nour Mourra, Shalom Sabwa, Marie M. Deschamps, Daniel W. Fitzgerald, Jean W. Pape, Denis Nash, Margaret L. McNairy

**Affiliations:** 1Department of Epidemiology and Biostatistics, Graduate School of Public Health and Health Policy, City University of New York, New York, NY 10017, USA; 2Institute of Implementation Science in Population Health, City University of New York, New York, NY 10027, USA; 3Haitian Group for the Study of Kaposi’s Sarcoma and Opportunistic Infections (GHESKIO), 33 Boulevard Harry Truman, Port-au-Prince 6110, Haiti; 4Center for Global Health, Weill Cornell Medicine, 402 East 67th Street, New York, NY 10065, USA; 5Division of General Internal Medicine, Weill Cornell Medicine, 402 East 67th Street, New York, NY 10065, USA

**Keywords:** household food insecurity, obesity, overweight, double burden of malnutrition, under-nutrition, over-nutrition, Haiti

## Abstract

Haiti is one of the most food-insecure (FIS) nations in the world, with increasing rates of overweight and obesity. This study aimed to characterize FIS among households in urban Haiti and assess the relationship between FIS and body mass index (BMI) using enrollment data from the Haiti Cardiovascular Disease Cohort Study. FIS was characterized as no/low, moderate/high, and extreme based on the Household Food Security Scale. Multinomial logistic generalized estimating equations were used to evaluate the association between FIS categories and BMI, with obesity defined as BMI ≥ 30 kg/m^2^. Among 2972 participants, the prevalence of moderate/high FIS was 40.1% and extreme FIS was 43.7%. Those with extreme FIS had higher median age (41 vs. 38 years) and were less educated (secondary education: 11.6% vs. 20.3%) compared to those with no/low FIS. Although all FIS categories had high obesity prevalence, those with extreme FIS compared to no/low FIS (15.3% vs. 21.6%) had the lowest prevalence. Multivariable models showed an inverse relationship between FIS and obesity: moderate/high FIS (OR: 0.77, 95% CI: 0.56, 1.08) and extreme FIS (OR: 0.58, 95% CI: 0.42, 0.81) versus no/low FIS were associated with lower adjusted odds of obesity. We found high prevalence of extreme FIS in urban Haiti in a transitioning nutrition setting. The inverse relationship between extreme FIS and obesity needs to be further studied to reduce both FIS and obesity in this population.

## 1. Introduction

Haiti, home to approximately 11.5 million people in 2021, is the poorest country in the Caribbean and Latin American region and has been continuously afflicted by high levels of food insecurity (FIS) [1]. FIS, the lack of regular access to sufficient safe and nutritious food for healthy living, is initially indicated by first worrying about having enough food, and may progress to making dietary changes to stretch food, and lastly, to decreasing food consumption in adults first and then children [2,3]. In 2021, over 80% of Haitians were estimated as facing moderate-to-severe FIS, and almost half were experiencing acute or severe FIS. Approximately 37% of Haitians live in the urban capital of Port-au-Prince. Urban areas had more young adults aged 20–39 years (35.5% vs. 28.8%) and more females (66.3% vs. 54.8%) than the total population in 2017 [4]. Education levels in urban Haiti are low, with an estimated 5% of adults completing secondary education [4]. Food insecurity is rapidly increasing in Haiti, from an estimated 29% of residents in the capital in 2016 [5], to current estimates from October 2022 indicating that Haitian residents now face “catastrophic” food insecurity—a designation that is a first for Haiti and for any country in the Western Hemisphere [6]. Reasons for FIS in Haiti are complex and multifactorial, spanning issues such as lack of food availability, access, and stability, due to the historical lack of sustainable development from geopolitical and environmental factors over several decades [7,8]. From its independence, Haiti’s infrastructure development was hampered by an unfair debt to France, supported by exploitive policies in Western countries [9]. In the 20th century, US occupation, coupled with Haiti’s national government, did not invest in sustainable institutions, detracting from business investment and resulting in a loss of fertile agricultural lands and farming infrastructure. Moreover, food tariffs and commoditization of foods have led to an overdependence on expensive, often poor-quality food imports, reducing the access and availability of fresh and nutritious foods in Haiti [10,11]. Environmental factors, such as climate-driven disasters and conflict, deepen the already weak food system by disrupting local agricultural production. Deforestation also left the land vulnerable to soil erosion from weather events, making it difficult to have sustainable local food [12,13]. In 2019, Haiti had the third-highest climate risk index in the world [14]. Earthquakes in Port-au-Prince in 2010 and in the Tiburon Peninsula in 2021, as well as Hurricane Matthew in 2016 and Tropical Storm Grace, each led to increased FIS [15,16,17,18,19]. During the COVID-19 pandemic, one study estimated that 85.5% of Haitians were worried about not having enough to eat the following week [20]. In urban Port-au-Prince, rival gangs disrupt food access by blocking roadways and other critical systems providing humanitarian aid [18,21]. Structural poverty and wealth inequity also play a role, with an estimated 60% of Haitians living in poverty [22]. As a result, an average working person in Haiti would need to pay 35% of their daily income to eat a basic meal [23]. Although Haiti’s Global Hunger Index slightly improved from alarming to serious in 2021 [24], the fragile state of Haiti due to these factors result in food instability.

Many lower-income countries (LIC) are experiencing a double burden of malnutrition, in which undernutrition and obesity co-exist [25]. Almost half of the population of Haiti were undernourished, while an estimated 31.2% of adult women and 22.2% of adult men were obese in 2020 [24,26]. FIS may manifest as both low and high BMI. One meta-analysis demonstrated that FIS was associated with 15% higher odds of obesity (pooled odds ratio: 1.15, 95% CI 1.06–1.23); however, severe FIS was associated with undernutrition [27]. When foods are not available, FIS may lead to undernutrition and weight loss, while when foods are available, individuals may compensate by eating cheaper foods that are higher in salt, fat, and refined carbohydrates, leading to weight gain. FIS may also cause adverse psychological stress, which increases cardiovascular disease risk [28,29,30]. In LIC, where severe FIS is more prevalent, the FIS–obesity relationship has been inconsistent [30,31]. A study among individuals living with HIV in a rural area in Haiti found that those with severe FIS had a less diverse diet but did not have lower BMI status compared to those with low FIS [32]. Globalization of the food system has increased access to fast food and processed food in LIC, creating conditions for a nutritional and epidemiologic transition to higher prevalence of non-communicable diseases [33].

Few studies have evaluated the effect of FIS in an urban area of Haiti on malnutrition, despite its high prevalence. To address this gap, this study investigates the relationship between FIS and malnutrition (defined as underweight or overweight/obese) in Port-au-Prince, Haiti. We aim to (1) characterize FIS among households in a population-based cohort from Port-au-Prince and (2) evaluate the association between FIS and malnutrition. We hypothesized that individuals from households with moderate/high and extreme FIS have higher prevalence of malnutrition compared to those with low FIS.

## 2. Materials and Methods

### 2.1. Data Source and Study Sample

A cross-sectional study using baseline data from the Haiti Cardiovascular Disease (CVD) Cohort Study, a population-based cohort of Port-au-Prince residents, was conducted [34]. The Haiti CVD Cohort used a multistage random sampling design to randomly select participants in Port-au-Prince using GPS waypoints across census blocks, with the number of waypoints per block proportional to the estimated population size [35]. Participants were eligible if they were >18 years old, had a primary residence in Port-au-Prince, and did not have any serious medical condition or cognitive impairment preventing participation. The study was conducted at the Groupe Haitien d’Etude du Sarcome de Kaposi et des Infections Opportunistes clinics (GHESKIO), a medical organization in Haiti which has provided clinical care for over 40 years and conducts research on HIV and chronic diseases. For this study, baseline data from participants enrolled between 15 March 2019 and 23 August 2021 were included. Participants who were pregnant were excluded (*n* = 33) (Appendix A).

### 2.2. Measures

#### 2.2.1. Household Food Insecurity

FIS was measured at the household level using an adapted version of the Six-Item Short Form of the Household Food Security Scale (study Cronbach’s alpha = 0.76), which measures FIS and hunger within the last 12 months and has been used in other LICs [36,37,38,39,40]. The scale consists of six questions indicating frequency of each FIS item. Traditionally, it is scored by assigning a point based on item responses of “often”, “sometimes”, “yes”, “almost every month” and “some months but not every month”, and then it is summed (range: 0–6) and categorized into high or marginal food security (0–1), low food security (2–4) and very low food security (5–6). However, given the high level of FIS in Haiti, an extreme FIS score was defined to capture differences across a population where the majority live with chronic food insecurity. The scoring was modified by assigning a point only to responses of “often”, “yes”, or “almost every month” on each item and then summing across items (range: 0–6). The score was then categorized as no/low FIS (0–2), moderate/high FIS (3–4), or extreme FIS (5–6). Classification of participants based on the original and the modified categories is found in Appendix A.

#### 2.2.2. Malnutrition

Malnutrition was defined as underweight, overweight, and obese based on the WHO classification using body mass index (BMI) categories [41]. BMI in kg/m^2^ was calculated using height and weight measured in the GHESKIO clinic at the baseline visit and classified as underweight: <18.5, normal: 18.5–<25, overweight: 25–<30, and obese: ≥30.

#### 2.2.3. Other Study Measures

Socio-demographic factors were measured on the individual level and included age, sex, education level, and individual income. Additional factors measured at the household level included total number of individuals living together. Family or neighborhood factors measured were perceived social support, social cohesion, and neighborhood violence. Perceived social support was based on the 12-item Multidimensional Scale of Perceived Social Support (MSPSS) [42]. Each item was scaled from 1: very strongly disagree-7: very strongly agree and scored as the mean of all items (range: 1–7). Social cohesion was based on the 5-item Collective Efficacy scale. Responses were coded as 1: strongly agree to 5: strongly disagree. After reverse-coding negative responses, item responses were summed to create a social cohesion score (range = 5–25). Neighborhood violence was based on the 5-item City Stress Index (CSI) [43]. Each response was scaled from 1: Never to 4: Often and the mean of all the items was calculated to determine a score (range = 1–4). An increase in perceived social support score, social cohesion score, and neighborhood violence score represented more social support, more social cohesion, or more neighborhood violence, respectively.

### 2.3. Statistical Analyses and Model Specification

Study characteristics were summarized by FIS categories. Unadjusted and multivariable multinomial logistic generalized estimating equations were fitted to evaluate population-averaged effects of FIS on BMI categories, using normal BMI as the referent outcome category. The multivariable model was adjusted for common variables associated with FIS and BMI (age, sex, household size, individual income, social support, neighborhood violence, and social cohesion) determined from the literature and guided using a directed acyclical graph (Appendix A). Continuous variables were mean-centered. Models accounted for clustering effects of households. Variance inflation factors < 5 for all variables, indicating that collinearity was not present. Due to missing data from BMI categories (*n* = 5, 0.1%), FIS (*n* = 62, 1.6%) and covariates (*n* = 33, 1.1%), complete cases (96.6%) were modeled. Unadjusted (OR) and adjusted odds ratios (aOR) and 95% confidence intervals (CI) were reported.

Several sensitivity analyses were performed. Potential effect measure modification between households with and without children, sex, and age group were also assessed by including statistical interactions between the potential modifiers and FIS. These were chosen because the six-item FIS instrument has lower validity for households with children and because of potential heterogeneity in FIS and malnutrition by demographics. R, version 4.1.2 was used for all analyses [44].

### 2.4. Ethical Considerations

All participants provided written informed consent before enrollment. Community, school, and religious leaders, as well as GHESKIO’s Community Advisory Board, were consulted prior to beginning the study. The institutional review boards at Weill Cornell Medicine and GHESKIO in Haiti approved all study procedures and protocols.

## 3. Results

### 3.1. Prevalence of FIS

The study included 2972 cohort participants from 1981 households who completed the baseline visit. Participants had a median age of 40 years (interquartile range (IQR): 28, 55) (Table 1). Just over half were female (57.6%), unmarried (55.2%), or had no children in the household (53.3%). Few had more than a high school education (14.9%) or earned >USD 10/day (17.7%).

Using the modified FIS score, the prevalence of moderate/high FIS was 40.1% and extreme FIS was 43.7%. Within the past 12 months, the overwhelming majority indicated skipping a meal or cutting portion size (83.8%), eating less (86.7%) or not eating when hungry due to lack of money (88.1%) (Figure 1). About half of participants indicated that food often did not last (51.2%) and that they often could not afford balanced meals (49.2%). Compared to those with no/low FIS, those with extreme FIS were on average older (median (IQR): 41 (28, 55) vs. 38 (27, 52) years) and less educated (secondary education: 11.6% vs. 20.3%).

### 3.2. Associations between FIS and Malnutrition

Participants with extreme FIS compared to those with no/low FIS had similar underweight prevalence (7.5% vs. 6.2%) and were less frequently obese (15.3% vs. 21.6%) (Figure 2). For participants with extreme FIS (*n* = 1299 (43.7%)), those who were obese (*n* = 198) compared to underweight (*n* = 97) were on average older (median (IQR): 46 (24–52) vs. 33 (24–52)) and had substantially higher proportions of females (92.9% vs. 46.4%). While still the majority, obese persons had lower proportions of earning no or little income (<USD 10/day) compared with those who were underweight (71.7% vs. 90.7%).

From the multivariable models shown in Table 2, there was an inverse dose response relationship, with moderate/high FIS (OR: 0.77, 95% CI: 0.56, 1.08) and extreme FIS vs. no/low FIS (OR: 0.58, 95% CI: 0.42, 0.81) associated with decreasing odds of being obese vs. normal weight. No association between FIS categories and being underweight were found. These effects were not modified by having children in the household, participant sex, or age (interaction *p* values > 0.10).

## 4. Discussion

Our study fills an important knowledge gap in the relationship between food insecurity and malnutrition, measured as underweight, overweight, and obese based on BMI categories, using a large community-based cohort in Port-au-Prince, Haiti. We found that the prevalence of moderate/high or extreme FIS was extremely high at 83.8% and that both moderate/high and extreme FIS compared to low FIS were associated with lower odds of being obese versus normal weight in a dose–response fashion, but were not associated with underweight versus normal weight.

The prevalence of moderate/high or extreme FIS in Port-au-Prince found in this study was similar to Haiti’s 2021 national estimate of >80%, though it was measured using different scales [45]. Although such high prevalence (72–95%) of moderate or severe FIS had been documented among vulnerable groups such as rural residents, pregnant women, and those living with HIV, [32,46,47], urban areas historically had lower prevalence of FIS [5,48]. However, a 2021 report noted a change in acute FIS during 2018–2021 from minimal to crisis designation in Port-au-Prince [49]. This alarming shift was likely the result of the 2021 earthquake and political instability during this period. After these events, rival gangs blocked major roads into Port-au-Prince and inflation of staple foods increased, worsening both food availability and food access [18,49].

Consistent with prior studies, the prevalence of moderate/high and extreme FIS was higher among those with lower education [32]. Although younger age was sometimes noted as a risk factor for FIS, in our sample, extreme FIS was more common among older adults [20]. Older participants may be especially vulnerable to FIS and its impact, possibly due to a higher likelihood of living alone without social support, and a greater difficulty accessing transportation to go to markets [50]. Although gender was correlated with FIS in other LIC, we did not find this association in our sample, and differences were previously found in rural communities among female farmers and widows and may not generalize to all women or women in urban settings [48,51]. The high overall prevalence of FIS in this region may also obscure smaller differences between subgroups.

While FIS has recently been associated in LICs with overweight/obesity due to excess consumption of low-quality calories from refined sugars and carbohydrates, facilitated by increased availability and access to cheap, edible oils and excess empty calories [33], we found the opposite association of extreme FIS associated with lower prevalence of obesity. In sub-Saharan Africa, high prevalence of overweight status (15.8%) was found among participants with FIS [31]. Severe FIS has also been associated with 30% higher odds for fast-food consumption in other LICs [52]. Our study found a lack of association between FIS and being underweight in a general urban population and builds on past studies showing this lack of association in special populations like persons living with HIV in rural Haiti [32].

A simple relationship between FIS, underweight, and overweight/obese status may not exist across multiple contexts, as caloric intake may change over time across populations and even within the same person due to fluctuating income, food availability, and access. Consumption of caloric-rich Western foods are suggested as the mechanism between FIS and obesity [30]. In Haiti, a transition to a Western diet has been noted, as well as an increase in obesity prevalence and a decrease in underweight prevalence [26,53]. Both those with extreme FIS and low FIS likely consume these foods, but not with the same regularity. The extreme FIS, who are extremely poor, may live in homes with inadequate means to cook regularly, such as indoor stoves, and may rely on prepackaged, highly processed foods rich in carbohydrates when available. However, they may not eat consistently due to frequent periods of reduced food availability and access. Neighborhood safety and lack of transportation in poor communities limit access to markets, street vendors, and restaurants selling nutrient-dense foods. Further, this vulnerable group is disproportionately affected by instability of the food supply due to geopolitical and environmental factors. When food can be purchased, there may be increased empty caloric intake, but when food availability and access are disrupted, those with extreme FIS go hungry, bounding their obesity prevalence. Those with low FIS, however, have more means to regularly consume high-caloric foods, leading to higher BMI. One study showed that Haitians who frequently ate high-caloric diets also had higher income and higher educational status [53]. Increased income was also associated with increased obesity prevalence in this cohort and those who were obese were more likely to eat from a street vendor [54]. Further, although those with extreme FIS may be less obese on average, they may still be at risk for cardiometabolic complications [55,56,57]. There was a 20% increase in cardiovascular disease (CVD) in Haiti from 2009–2019 and CVD was the leading cause of death in 2019 [58]. Focusing on BMI categories to reduce CVD may exclude this vulnerable group.

Both long-term and short-term resilience to food insecurity is needed. Multilateral organizations, in coordination with the government of Haiti, have adopted a systems approach to improve food security with long-term policy goals [12,13,59]. Short-term interventions include emergency aid programs, increased screening within clinics and schools, and increased health-care provider training on malnutrition treatment. Longer-term interventions must strengthen the education, healthcare systems, and supply of local agriculture in Haiti. In addition, larger structural interventions that encourage and incentivize local agriculture sector growth of local nutritious foods that are more affordable than expensive exports are needed. Food insecurity is also increasing due to climate change [60]. Programs for local and safe food storage and food distribution plans may provide resilience to interruptions to the food supply from natural disasters [60]. Climate-smart agriculture, such as water retention plans, hillside terracing, greenhouses, and food gardens should expand to urban areas [11,61]. These potential local sources of food may be resistant to internal obstacles of food delivery. Additionally, novel community-based models of care should be considered, including Haiti’s success with using community health workers to provide education, screening, and food distribution. Community health workers have been used by GHESKIO for over 20 years during public health epidemics to provide emergency cholera vaccination, tuberculosis screening, and malnutrition screening. Data specifically from our study found that policies should target older adults and those with low education, as they were more vulnerable to extreme FIS. Examples of policies or interventions could include public media and education campaigns, primary-school-based food insecurity assessment and aid, and community-organization-based education and aid programs targeting these specific groups.

By quantifying FIS, as well as identifying groups vulnerable to extreme FIS in urban Haiti, this study may support policy-makers in designing interventions and allocating resources to address FIS. This study also highlights the complexity of the nutritional transition in Haiti by evaluating the relationship between FIS and malnutrition. However, some limitations of our work should be considered. First, the six-item Household Food Security Scale from the last 12 months was not validated in Haiti, and measurement from other studies using the Household Hunger Scale, which prompted responses within the last 4 months, may contribute to prevalence differences; however, our modified scale was restrictive, and most participants endorsed food-insecure behaviors occurring often and almost every month. The six-item scale also did not directly ask about child hunger, which is an indication of the most severe household food insecurity. The sensitivity analyses, however, did not demonstrate differences in effects by the presence of children in the household. Second, results from an urban area of Haiti may not be generalizable to other populations in Haiti with different socio-economic status, higher income, or to other time periods given the influential events in 2019–2021. Third, BMI has been shown to misclassify participants as overweight or obese [62]. Future analyses using other anthropomorphic measures such as the waist-to-hip or waist-to-height ratio should be considered [63]. Lastly, since this is a cross-sectional study, no temporal or causal relationships can be inferred.

## 5. Conclusions

In summary, we found high proportions of extreme FIS in an urban region of Haiti, which is higher among the less-educated and older adults. Extreme FIS is a metric that may be a useful in identifying the most vulnerable and in need of immediate interventions in extremely low-income countries with large prevalence of food insecurity. The inverse relationship between FIS and obesity underscores the complexity of malnutrition, suggesting that policies and interventions are needed to confront FIS and obesity as dual health targets. Haiti and similar low-income countries facing dual challenges of FIS and obesity need immediate and long-term interventions, ranging from FIS and obesity screening programs with direct food subsidies to longer-term plans for development of agricultural infrastructure, increased education, and food safety nets. As the ongoing food insecurity crisis in Haiti continues, these interventions to improve the resilience to food insecurity, as well as studies evaluating their effectiveness, are paramount for improving health outcomes across the region.

## Figures and Tables

**Figure 1 nutrients-14-04854-f001:**
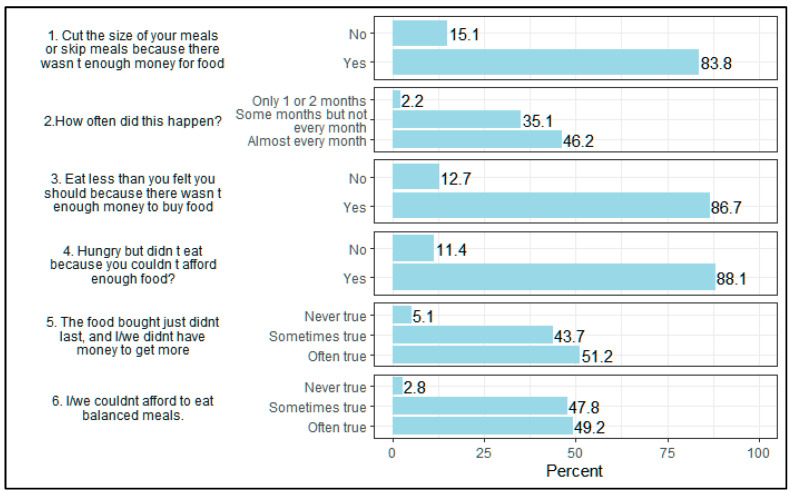
Household Food Insecurity Item Responses from the Haiti CVD Cohort Study, 2019–2021 (N = 2972) Items with responses of unknown, refused, or missing were not presented and percent may not total 100%.

**Figure 2 nutrients-14-04854-f002:**
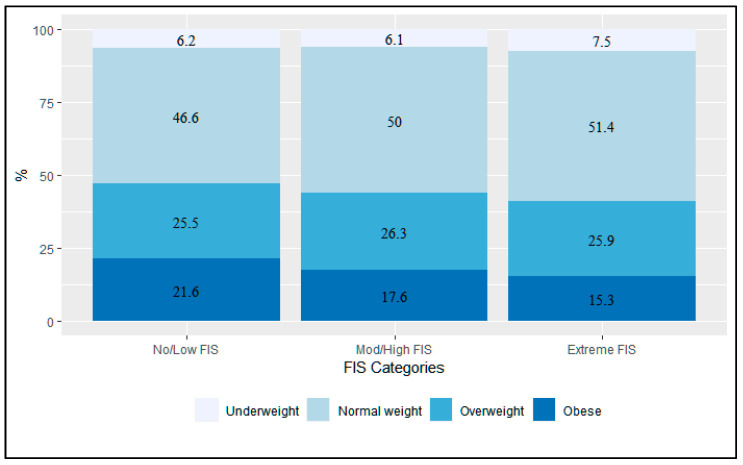
Prevalence of malnutrition by FIS from the Haiti CVD Cohort Study, 2019–2021 (N = 2972).

**Table 1 nutrients-14-04854-t001:** Characteristics compared by food insecurity categories from the Haiti CVD Cohort Study, 2019–2021 (*n* = 2972).

Characteristic	No/Low FIS	Mod/High FIS	Extreme FIS	Overall ^a^
Total	418 (14.1)	1193 (40.1)	1299 (43.7)	2972 (100)
Age in years, Med (IQR), (Range)	38 (27, 52), (18, 93)	41 (28, 55), (18, 90)	41 (28, 55), (18, 90)	40 (28, 55), (18, 93)
Age group, n (%)				
≤39 years	221 (52.9%)	557 (46.7%)	618 (47.6%)	1430 (48.1%)
40–59 years	138 (33.0)	430 (36.0)	445 (34.3)	1028 (34.6)
60+ years	59 (14.1)	206 (17.3)	236 (18.2)	514 (17.3)
Female Sex, n (%)	251 (60.0)	661 (55.4)	766 (59.0)	1712 (57.6)
Highest level of school, n (%)				
≤Primary	121 (29.0)	418 (35.0)	503 (38.7)	1066 (35.9)
Secondary	211 (50.5)	576 (48.3)	644 (49.6)	1457 (49.0)
Higher than secondary	85 (20.3)	194 (16.3)	151 (11.6)	442 (14.9)
Missing	1 (0.2)	5 (0.4)	1 (0.1)	7 (0.2)
Current marital status, n (%)				
Single	231 (55.3)	648 (54.3)	729 (56.1)	1642 (55.2)
Living together	61 (14.6)	182 (15.3)	244 (18.8)	493 (16.6)
Married	100 (23.9)	291 (24.4)	249 (19.2)	661 (22.2)
Widowed/Divorced/Separated	25 (6.0)	67 (5.6)	76 (5.8)	169 (5.7)
Missing	1 (0.2)	5 (0.4)	1 (0.1)	7 (0.2)
Income, n (%)				
Less than USD 10/day	342 (81.8)	980 (82.1)	1065 (82.0)	2440 (82.1)
More than USD 10/day	75 (17.9)	208 (17.4)	233 (17.9)	525 (17.7)
Missing	1 (0.2)	5 (0.4)	1 (0.1)	7 (0.2)
No. people currently live in household, Med (IQR), (Range)	3 (2, 5), (1–13)	3 (2, 5), (1–14)	3 (2, 5), (1–15)	3 (2, 5), (1–15)
Perceives social support score, Med (IQR), (Range)	5.64 (4.55, 6.45), (1, 7)	5.55 (4.45, 6.45), (1, 7)	5.55 (4.36, 6.52), (1, 7)	5.55 (4.45, 6.45), (1, 7)
Missing	3 (0.7)	5 (0.4)	1 (0.1)	9 (0.3)
Social cohesion score, Med (IQR), (Range)	15 (14, 16), (7, 22)	15 (13, 16), (7, 24)	15 (13, 16), (7, 24)	15 (13, 16), (7, 24)
Missing	1 (0.2)	5 (0.4)	1 (0.1)	7 (0.2)
Neighborhood violence score, Med (IQR), (Range)	9 (7, 11), (5–17)	9 (7, 10), (5–17)	9 (7, 11), (5–20)	9 (7, 11), (5–20)
Missing	2 (0.5)	16 (1.3)	14 (1.1)	32 (1.1)

CVD = cardiovascular disease; FIS = household food insecurity; Med = median; IQR = interquartile; HH = household. ^a^ FIS values missing for 62 participants.

**Table 2 nutrients-14-04854-t002:** Association between FIS categories ^a^ and malnutrition ^b^, from multinomial logistic general estimating equations.

BMI Categories	Unadjusted OR (95% CI)	Adjusted OR (95% CI)
Underweight vs. Normal		
No/Low FIS	Ref	Ref
Mod/High FIS	0.90 (0.56, 1.44)	0.96 (0.60, 1.55)
Extreme FIS	1.07 (0.68, 1.69)	1.12 (0.70, 1.78)
Overweight vs. Normal		
No/Low FIS	Ref	Ref
Mod/High FIS	0.97 (0.74, 1.26)	0.94 (0.71, 1.24)
Extreme FIS	0.93 (0.71, 1.21)	0.88 (0.67, 1.16)
Obese vs. Normal		
No/Low FIS	Ref	Ref
Mod/High FIS	0.77 (0.57, 1.04)	0.77 (0.56, 1.08)
Extreme FIS	0.65 (0.48, 0.87)	0.58 (0.42, 0.81)

BMI = body mass index; FIS = food insecurity; OR = odds ratio; CI = confidence interval; Ref = reference level. ^a^ FIS categories based on the modified version of the Six-Item Short Form of the Household Food Security Scale. No/low FIS (0–2), moderate/high FIS (3–4), and extreme FIS (5–6). ^b^ Estimates based on unadjusted and multivariable multinomial logistic general estimating equations of the effects of FIS on BMI categories, using normal BMI as the reference outcome. The multivariable model adjusted for age, sex, household size, income, social support, neighborhood violence, and social cohesion.

## Data Availability

Data contain potentially identifying and sensitive patient information. Deidentified data used for this analysis are available upon request after signing a data access and use agreement, provision of approval by the GHESKIO ethics board, and demonstration that the external investigative team is qualified and has documented evidence of human research protection training. Requests may be addressed to the authors or to irb@med.cornell.edu.

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
