# Peer review of "Extreme Food Insecurity and Malnutrition in Haiti: Findings from a Population-Based Cohort in Port-au-Prince, Haiti"

_nutrients, 2022, doi:10.3390/nu14224854_

Round 1

Reviewer 1 Report

The study presents a topic of interest by showing how food insecurity affects a poor Latin American country, generating implications associated with obesity, malnutrition and poverty.  The importance of this study should not be limited to pointing out the existence of food vulnerability in the population but to generate strategies and public policies that contribute to its mitigation. Other countries (e.g., Honduras, El Salvador, Paraguay) in the region suffer from food insecurity and their governments are unable to contain this phenomenon.

The authors define what food insecurity is, but do not explain how food insecurity emerged in Haiti and how it was configured. Is it important to have a historical vision to understand the problem, although the authors point out some elements associated with climate change and the absence of food policies is still insufficient, the lack of resilience in the population is perceived, why does it happen?

The methodological section is logical and coherent and explains in detail the research design, an important aspect for replication. The results and discussion sections are sufficiently explained based on the methodology used in the study.

The conclusions are poor and insufficient, and need to be re-elaborated. The results have sufficient elements to conclude not only on the research objective but also on the scenary of the problem in the country and in the region.

I recommend the authors to expand the theoretical basis of the study, which I pointed out at the beginning of my remarks. On the other hand, it is no secret that Haiti is a country with extreme poverty and a high rate of malnutrition, this study and others show it, but we do not need more studies that do not say that, because we already know that, solutions to the problems presented are required, the authors point out several elements that can guide policy makers, but they do not develop this aspect that is of utmost importance for Latin America and the Caribbean, For example, they observed a correlation between the schooling of individuals and food insecurity. Perhaps a public policy aimed at remedying this situation would have a positive impact on the region through, for example, an improvement in educational infrastructure, food programmes with scholarships, educational programmes in rural areas, and assistance from multilateral organisations.

This last aspect may be the reason why the Haitian territory has little individual resilient response to natural disasters; it is a line of enquiry that needs to be explored or recommended in a possible research agenda.  The study is interesting and it would be a valuable contribution if my recommendations were included.

Reviewer 2 Report

I am grateful for the opportunity to review the manuscript presented to me.

I hope that the comments in the review would be helpful. I believe the paper is worth considering for publication, however requires minor revision.

All comments are in the attached manuscript in review mode

Round 2

Reviewer 1 Report

The submitted manuscript has slight improvements, the authors do not consider my recommendations, I suggest not to publish the manuscript until the authors address the observations mentioned in the first revision.
